

# The AROME-WMED re-analyses of the first Special Observation Period of the Hydrological cycle in the Mediterranean experiment.

Nadia Fourrié[1], Mathieu Nuret[1], Pierre Brousseau[1], Olivier Caumont[1], Alexis Doerenbecher[1], Eric Wattrelot[1], Patrick Moll[1], Hervé Bénichou[2], Dominique Puech[1], Olivier Bock[3], Pierre Bosser[4], Patrick Chazette[5], Cyrille Flamant[6], Paolo Di Girolamo[7], Evelyne Richard[8], and Frédérique Saïd[8]

[1]CNRM, Université de Toulouse, Météo-France, CNRS, Toulouse, France
[2]Météo-France, Toulouse, France
[3]IGN, Univ. Paris Diderot, Paris, France
[4]ENSTA Bretagne - Lab-STICC UMR CNRS 6285 - PRASYS Team, Brest, France
[5]LSCE, Gif sur Yvette, France
[6]Laboratoire Atmosphères Milieux Observations Spatiales, Sorbonne Université, Université Paris-Saclay and CNRS, Paris, France
[7]Scuola di Ingegneria, Università della Basilicata, Italy
[8]Laboratoire d'Aérologie, Université de Toulouse, CNRS, UPS, Toulouse, France

**Correspondence:** Nadia Fourrié (nadia.fourrie@meteo.fr)

**Abstract.** To study key processes of the water cycle, two special observation periods (SOPs) of the Hydrological cycle in the Mediterranean experiment (HyMeX) took place during the autumn 2012 and winter 2013. The first SOP aimed to study high precipitation systems and flash-flooding in the Mediterranean area. The AROME-WMED (West-Mediterranean) model (Fourrié et al., 2015) is a dedicated version of the mesoscale Numerical Weather Prediction (NWP) AROME-France model

which covers the western Mediterranean basin providing the HyMeX operational centre with daily real-time analyses and forecasts. These products allowed adequate decision-making for the field campaign observation deployment and the instrument operation. Shortly after the end of the campaign, a first re-analysis with more observations was performed with the first SOP operational software. An ensuing comprehensive second re-analysis of the first SOP which included field research observations (not assimilated in real-time), and some reprocessed observation datasets, was made with AROME-WMED. Moreover, a more

recent version of the AROME model was used with updated background error statistics for the assimilation process.

This paper depicts the main differences between the real-time version and the benefits brought by HyMeX re-analyses with AROME-WMED. The first re-analysis used 9% of additional data and the second one 24% more compared to the real-time version. The second re-analysis is found to be closer to observations than the previous AROME-WMED analyses. The second re-analysis forecast errors of surface parameters are reduced up to the 18-h or 24-h forecast range. In the mid and in the upper-

15 troposphere, upper-level fields are also improved up to the 48-h forecast range when compared to radiosondes. Integrated Water Vapour comparisons indicate a positive benefit for at least 24 hours. Precipitation forecasts are found to be improved with the second re-analysis for a thresholds up to 10 mm/24-h. For higher thresholds, the frequency bias is degraded. Finally, improvement brought by the second re-analysis is illustrated with the Intensive Observation Period (IOP 8) associated with heavy precipitation over Eastern Spain and South of France.



## 1 Introduction

The HYdrological cycle in the Mediterranean EXperiment (HyMeX, Drobinski et al. (2014)) is a ten-year scientific programme aiming at a better understanding and quantification of the hydrological cycle and related processes in the Mediterranean region. An emphasis is given on high-impact weather events, inter-annual to decennial variability of the Mediterranean coupled system, and associated trends in the context of global climate change. The first special observing period took place in Autumn 2012 (05 September to 06 November 2012) to study the heavy precipitation and flash flooding events (Ducrocq et al., 2014).

An AROME (Application of Reasearch to Operations at Mesoscale, (Seity et al., 2011)) model version dedicated to the HyMeX programme, AROME-WMED (West Mediterranean) model (Fourrié et al., 2015) centred over the western Mediterranean basin, was developed in 2009 to study heavy precipitation in this region. Several studies have indeed shown the importance of an accurate description of the low-level moist flow feeding mesoscale convective systems, which can result in heavy precipitation events (Duffourg and Ducrocq, 2011; Bresson et al., 2012; Ricard et al., 2012). During the HyMeX Special Observing periods, a real-time version of AROME-WMED model (Fourrié et al., 2015) with data assimilation, called hereafter SOP1, was run to provide scientists with analyses and forecasts of meteorological situations. These forecast fields were also used to drive ocean and hydrological models and allow the guidance for observation deployment planning and safety management of the observation platforms and the instruments.

During the campaign, innovative observations came from boundary layer pressurized balloons (BLPBs) (Doerenbecher et al., 2016) developed by CNES (Centre National d'Etudes Spatiales) and airborne in situ and remote sensing observations from the French SAFIRE Falcon 20 and ATR-42 and the German Dornier aircraft. Radiosondes were also launched from mobile platforms along the French and Italian Mediterranean coasts and in Corsica depending on meteorological situations. Moreover, additional operational radiosondes were activated on request at 06:00 and 18:00 UTC through the Data Targeting System (DTS) implemented by ECMWF (European Centre for Medium-range Weather Forecasts; Prates et al. (2009)) within the EUMETNET Observation Programme.

In the past, several re-analyses were performed after experimental campaigns such as for the Fronts and Atlantic Storm-Track EXperiment (Desroziers et al., 2003) or the Mesoscale Alpine Programme (Keil and Cardinali, 2004) with a view to provide a new reference description for process studies. In the frame of the Innovative Observing and Data Assimilation Systems for severe weather events in the Mediterranean project, it was decided to perform re-analyses of the HyMeX Special Observing Period to benefit from additional research observations, as well as from advances in assimilation algorithms and modelling.

Shortly after the HyMeX campaign, a first re-analysis (REANA1), which did not include any new data processing, was performed to provide scientists with an unified dataset for process studies. The real-time AROME-WMED version was indeed upgraded during the field campaign on 25 September 2012 at 06 UTC. More recently, a second re-analysis of the HyMeX special observation period (REANA2) was undertaken to take advantage of observations deployed during the field campaign not included in SOP1 and REANA1 as well as enhanced reprocessed datasets. REANA2 also benefited from progress from the lastest model developments.



|  | SOP1 | REANA1 | REANA2 |
|---|---|---|---|
| Lateral boundary conditions | ARPEGE cy36+cy37 | ARPEGE cy37 | ARPEGE cy37 |
| Topography | GTOPO30 | GTOPO30 | GMTED2010 |
| Background errors | estimated from a 2010 period | estimated from a 2010 period | estimated from 15 day period |
|  |  | ( | from 17 to 31 october 2012) |

**Table 1.** AROME-WMED re-analysis and model configurations (main differences): SOP1 for real-time, REANA1 for first re-analysis, REANA2 for second renalysis.

The aim of this paper is to review the main characteristics of the AROME-WMED re-analysis versions in terms of data assimilation and forecast and to compare them with their real-time counterpart. The outline of the paper is as follows. Section 2 compares both configurations of the AROME-WMED re-analysis and the real-time versions The different datasets assimilated in the re-analyses are specified in section 3. Section 4 evaluates the assimilation and forecast with respect to various observations. The qualitative and quantitative precipitation evaluation of the three AROME-WMED versions for the Intensive Observation Period (IOP) 8 case study is discussed in section 5. Conclusions are found in section 6.

## 2 Description of the AROME-WMED model

### 2.1 Model configurations

The AROME-WMED model strongly relies on the AROME-France model, which is the Météo-France operational limited-area model (Seity et al., 2011; Brousseau et al., 2016). This model is based on a non-hydrostatic equation system (Bénard et al., 2010). At the time of the campaign (2012), it had a $2.5 \times 2.5$ km horizontal mesh and 60 vertical levels ranging from 10 m above the surface to 1 hPa. A one-moment microphysical parametrisation (Pinty and Jabouille, 1998; Caniaux et al., 1994), which takes into account five classes of hydrometeors (cloud liquid water, cloud ice, rain, snow and graupel) is used. Two schemes represent the vertical turbulent transport in the boundary layer: an eddy diffusivity scheme based on a prognostic turbulent kinetic energy parameterization (Cuxart et al., 2000) and a mass flux scheme (Pergaud et al., 2009) to account for dry thermal and shallow convection. There is no deep convection parametrisation. A specific algorithm named CANOPY (Masson and Seity, 2009) diagnoses the 2-m temperature, 2-m humidity and 10-m wind at every time step in the surface scheme (SURFEX; Masson et al. (2013)).

The AROME-WMED domain (34N 11W, 48N 20E) ranges from Portugal to Italy and from North Africa to France (Figure 1). It was designed to study high-precipitation events which occur along the north western Mediterranean coasts, from Catalonia to Central Italy. The model grid includes a $960 \times 640$ point matrix, centred on 41.5N, 4.1E.

Table 1 lists the main differences of configuration in the model. The same method as in AROME-France was used to set up the surface characteristics for the SURFEX scheme. Physiographic data are initialized over the AROME-WMED domain using the so-called ECOCLIMAP database at 1km resolution (Masson et al., 2003). The topography is extracted from the





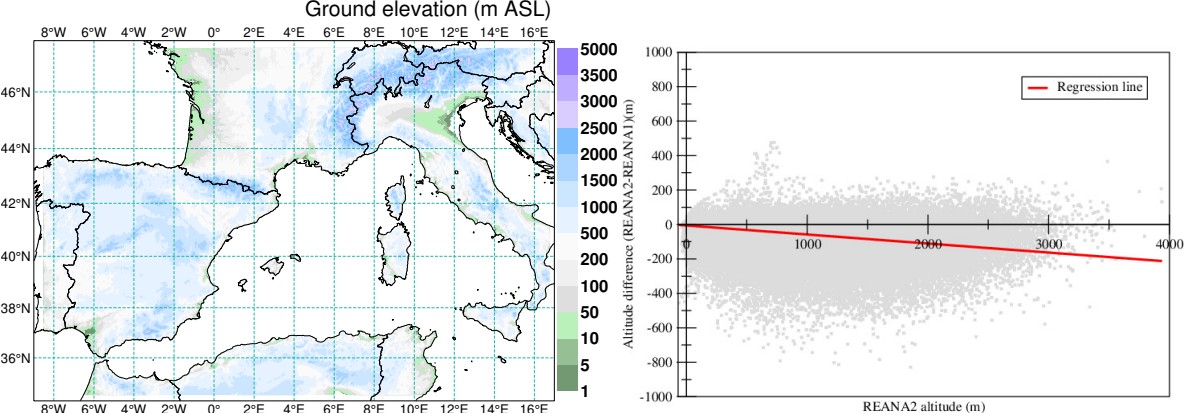

**Figure 1.** REANA2 orography (left panel) and difference between REANA2 vs REANA1 (right panel). The red line corresponds to the regression between both dataset.

Global 30 Arc-Second Elevation Data Set (GTOPO30, http://eros.usgs.gov/products/elevation/gtopo30.html) database for the real-time version and the first re-analysis. In the second re-analysis, the Global Multi-resolution Terrain Elevation Data 2010 (GMTED2010, Danielson and Gesch (2011)) database was used. A mean difference of -21 m was found between the REANA2 orography compared to the REANA1 and SOP1 versions (Fig. 1).

5    Lateral boundary conditions are provided by the Météo-France global NWP ARPEGE system (Courtier et al., 1991). For REANA2, ARPEGE forecasts benefit from a maximum of assimilated data using longer cut-off analyses than for REANA1 and SOP1. Once per day, a 54-h forecast is run at 00 UTC for both re-analyses compared to the 48-h forecast range of the real time version. This allows the comparison of 24-h forecasted precipitation with raingauges which are mainly available for the period 06 UTC-06UTC on the following day.

## 10  2.2   Data assimilation and background error statistics

Initial atmospheric states of AROME-WMED come from 3D-Var analyses. These analyses are performed every 3-h assimilating observations taken within a +/- 1h30 assimilation window. The first guess is the 3h forecast from the previous analysis time. The analysed parameters are the temperature, the specific humidity, the two horizontal components of the wind and the surface pressure. For the surface analysis, an optimal interpolation scheme is used to analyse soil temperature, soil humidity

15  over land and sea surface temperature from data measured with surface stations and buoy observations (Masson et al., 2013).

    The background error covariance matrix (the so-called B matrix) is a key component of the variational assimilation system, as it weighted the spread of the observation impact in the data assimilation system. As in AROME-France, a climatological background error covariance matrix is used and has been computed from an AROME-WMED data assimilation ensemble using the ensemble approach proposed by Brousseau et al. (2011). In the real time version and in the first re-analysis, the

20  background error covariance matrix was computed over a 1-week period in October 2010, characterized by convective systems over southern France and Catalonia.





**Figure 2.** Variance spectrum for specific humidity a), temperature b), divergence c) and vorticity d) for the SOP1 and REANA1 version (dashed black line) and REANA2 (blue line) at about 600 hPa.



**Figure 3.** Background error standard deviation for specific humidity a), temperature b), vorticity c) and divergence d) for the SOP1 and REANA1 versions (dashed black line) and REANA2 (blue line) at around 600 hPa.





For the second re-analysis, the background error covariance matrix was computed over a longer period of the HyMeX special obervation period (17 to 31 October 2012) : this new B matrix is more representative of the encountered meteorological conditions. Comparing the variance error spectra of both matrices, (see for example in figure 2 error variance spectra at around 600 hPa), it appears that for all parameters, the error variances for REANA2 are smaller for the smaller horizontal scales of the

model and on the contrary, are above for the larger ones, due to meteorological situations involving fewer small scale features than during the period in October 2010, used to estimate the B matrix for SOP1 and REANA1. These changes in variance spectra are twofold:

First, for temperature and specific humidity (resp. vorticity and divergence), this increase (resp. decrease) occurring for scales in the maximum of the variance spectra leads to a general increase (resp decrease) of spectrally averaged background

errors (figure 3) in the new B matrix. This means that using the same background and a same observation, the analysis fits better (resp. lesser) the temperature and humidity (resp. wind) observations using the REANA2 B matrix than the SOP1/REANA1 one.

Secondly, horizontal correlations length-scales are slightly longer which allows each observation to modify the analysis over a more horizontally extended area.

The other components of the background error covariances (i. e. vertical correlations and cross-correlations between the different analyzed model fields) are similar for both B matrices (not shown).

## 3   Assimilated data

### 3.1   Observations common to all AROME-WMED versions

Both REANA1 and REANA2 re-analyses used all available data with no time constraint (cut-off), contrary to the SOP1 (real-

time) version. These observations come from radiosondes, including mobile sites along the French Mediterranean coast, surface stations and buoys, aircraft and wind profilers. Satellite data are dominant in the analysis, contributing to more than 50% of the assimilated dataflow, since a large part of the domain is over the sea. Satellite data comprise infrared and microwave radiances from polar-orbiting satellites, radiances from SEVIRI on board Meteosat Second Generation (MSG), surface wind from scatterometers over the Mediterranean Sea and atmospheric motion vectors.

The GNSS (Global Navigation Satellite System) Zenith Total Delay (GNSS-ZTD) observations from the EUMETNET EIG GNSS water vapour programme (E-GVAP) network are assimilated as well. Another major data source is the French Doppler Radar network (around 18 radars in the AROME-WMED domain), which provides Doppler winds (Montmerle and Faccani, 2009) and reflectivities, from which are derived relative humidity profiles (Caumont et al. (2010); Wattrelot et al. (2014)), but their density is weather dependent, i.e. presence of rain or not. Fourrié et al. (2015) provide complementary information about

assimilated data.





|  | Assimilated Variables | REANA1 | REANA2 |
|---|---|---|---|
| GNSS | zenithal total delays | real-time version | reprocessed version (V3) |
| Radiosonde | T,q,u,v | low resolution (TEMP) + HYMEX mobile sites 01, 02 and 03 | high resolution (where available) + L'Aquila + Biscarosse + dropsondes |
| RADAR | radial wind, reflectivity | FRANCE | FRANCE+SPAIN |
| Research Aircraft | T,u,v | X | Falcon-20, ATR, Dornier |
| Water Vapour Lidar | q | X | Ground-based: BASIL and WALI Airborne: Leandre (ATR42) |
| Profiler | u,v | real-time version | reprocessed version |
| Boundary layer pressurized balloons | T, r, u,v | reprocessed data | reprocessed data only night-time for |

**Table 2.** Main differences, in terms of assimilated data between the first re-analysis (REANA1) and the second one (REANA2).

### 3.2 Observations specific to REANA2

In addition new dataset and reprocessed observations were assimilated in REANA2. Table 2 summarises the main differences in terms of assimilated observations between both re-analyses. The GNSS zenithal total delays from the reprocessed dataset available in the HyMeX database (Bock et al., 2016) have been used. The methodology for their assimilation is described in

Mahfouf et al. (2015). All available GNSS data were reprocessed homogeneously with a single software, more precise satellite orbits and clocks, and additional sites were taken into account (e.g. Sardinia). This led to a better coverage as shown in Fig. 4, especially over France, the Iberian Peninsula and Italy. Furthermore, an updated static bias correction for each couple (GNSS station, analysis centre) was computed for the REANA2 version. Data from BLPBs (temperature, humidity and wind) were assimilated in both re-analyses REANA1 and REANA2. The raw data were averaged on 20-minute period approximately.

Moreover, to guarantee the consistency of such data, averaging was only performed over periods corresponding to stabilized flight segments. In REANA2, temperature data were discarded during daytime due to radiative bias and model errors in the boundary layer.

    High vertical resolution radiosondes, where available (in France including dedicated HyMeX mobile soundings, in some sites in Spain, as shown in Fig. 4), were used, instead of the classical TEMP messages, assimilated in the SOP1 and REANA1

versions as proposed in Ingleby et al. (2016). This leads to an increased data flow (100 to 150 data per profile instead of 30 for the TEMP message); extra sounding sites were also processed, such as L'Aquila (a research center in Italy) and Biscarosse, a French military site close to the Atlantic coast. Data from several Spanish Doppler radars (Valencia, Barcelona, Murcia, Almeria and Palma) have also been used in the second re-analysis after a careful quality control. Wind profilers data have also been carefully checked in order to remove spurious signals (Saïd et al., 2016). Humidity retrieved from ground-based and

airborne lidars have been processed. Two ground-based research lidars were available: one located in Candillargues (BASIL instrument, Di Girolamo et al. (2016)) and the other one in Menorca Island - Spain (WALI instrument, Chazette et al. (2016)).





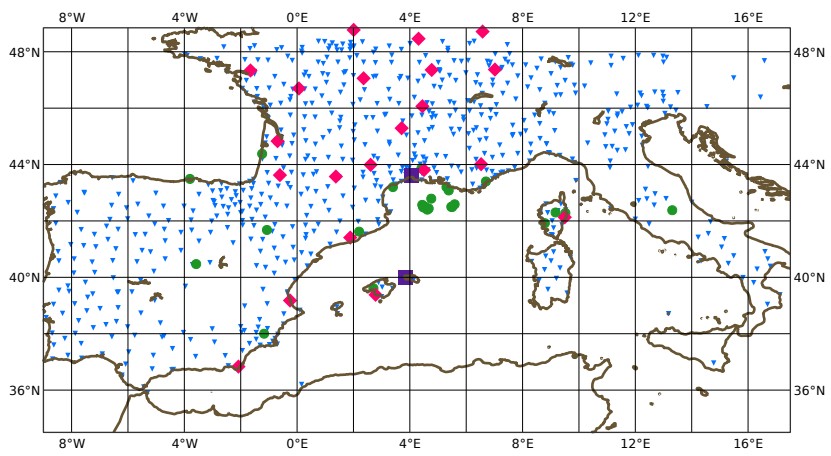

**Figure 4.** REANA2 assimilated data focus: green disks represent the location of radiosondes (fixed and mobile) taken into account at high resolution, red diamonds the position of the Doppler radars, violet squares the Lidar sites and blue triangles the GPS-GNSS stations.

These data have been smoothed at a 200 m vertical resolution and outliers have been removed. The lidar Leandre II data (temperature and wind) from 22 ATR flights were also assimilated according to the method described in Bielli et al. (2012); these data were thinned at a 15 km horizontal resolution to avoid horizontal error correlation problems in the data assimilation process.

5  The amount of assimilated data per observation type for each AROME-WMED analysis version is given in Fig. 5. The number of assimilated data in REANA1 (red bars) is slightly increased with respect to the SOP1 version (black bars). This can be explained by the fact that all available observations and not only those present in real-time in the Météo-France database were assimilated. +9% additional data were thus assimilated in REANA1 compared to SOP1. Concerning the REANA2 (blue bars), +24% additional data with respect to SOP1 and +13% with respect to REANA1 were assimilated. The igher observation

10  number mainly comes from radiosondes (higher resolution and additional sites), profilers, radiances, scatterometer winds, surface parameters and ground based GNSS data. However, although five Spanish Doppler radars were included in REANA2, less data from radars were assimilated as a consequence of a revised statistics tuning.

Examples of the assimilated data distribution for a rainy day (26 September 2012) and a non-rainy day (5 October 2012) are shown in Figure 6. First of all, satellite data contribute most to the observational set. This distribution varies depending on





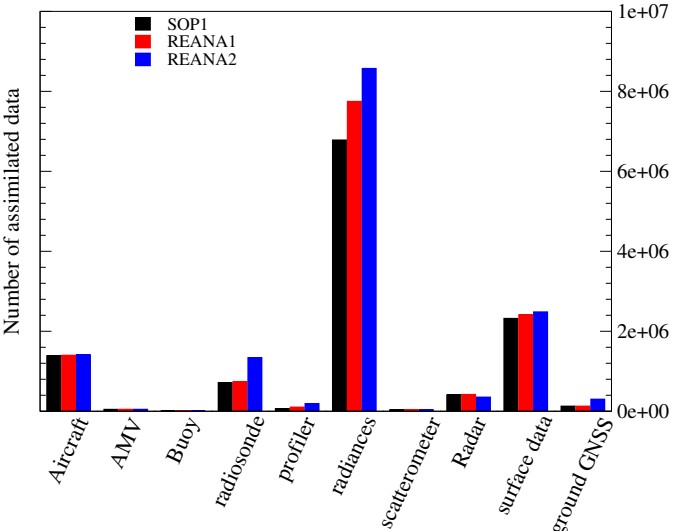

**Figure 5.** Number of assimilated data in the AROME-WMED model for the real-time version (SOP1), the first re-analysis (REANA1) and the second re-analysis (REANA2).

weather conditions (rainy/non-rainy). For the rainy day, radar data represent 6% of the total. The percentage of satellite data is reduced from 63.5% to 50% for a non rainy day. Infrared measurements (SEVIRI and IASI) are indeed strongly affected by the presence of clouds and thus discarded. In this case the proportion of radiosondes data increases for the rainy day (twice the amount of non-rainy day, due to additional radiosondes). The large increase in radiosonde data for 26 September 2012 is

5 explained by the fact that the DTS was activated resulting in an increased frequency of radiosonde launches at specific sites.

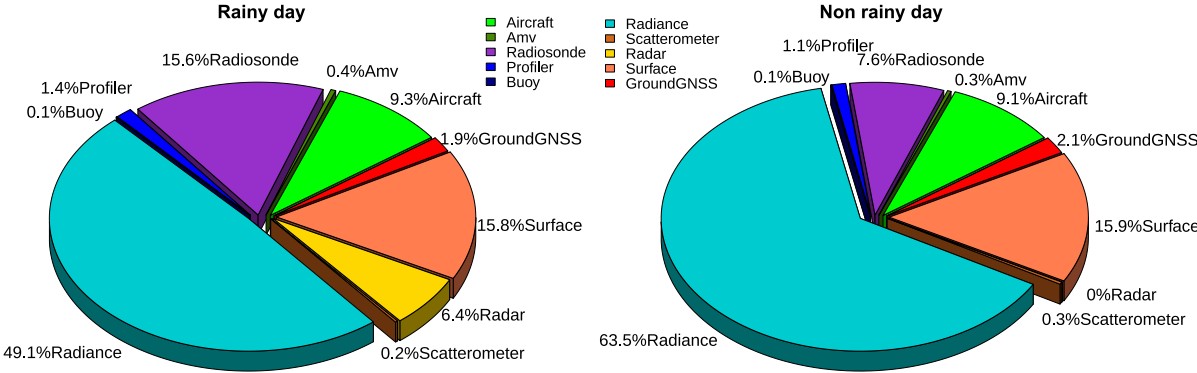

**Figure 6.** Distribution of assimilated data in the second re-analysis (REANA2) for 26 September 2012 (8 analysis times, left panel, rainy day) and for 05 October 2012 (right panel, non rainy day).





## 4    Assimilation results

### 4.1    Analysis and First-Guess

As a first validation step, the performances of the data assimilation system from the three AROME-WMED sets were evaluated based on the analysis (AN) and first-guess (FG is the 3h forecast) departures from the assimilated observations. These

departures provide information on the analysis increment for AN and on very short range forecast quality for FG. Some of these statistics (mean and RMS) are plotted on figure 7 (resp. figures 8 and 9) for observations related to humidity (resp. wind). These datasets differ with respect to the AROME-WMED version as the quality check based on the difference between the observation and the simulation can discard or not some observations due to a different background value. In addition some observations type such as Lidar observations or Spanish radars are specifically assimilated in REANA2. For the radiosondes

and the wind profilers, the real-time observations were replaced with high resolution data and reprocessed data respectively in REANA2.

First of all, for all observations types, the RMS of AN departures are always smaller than the corresponding FG departures as expected for a well-performing assimilation process.

As SOP1 and REANA1 use the same background statistics, results of these 2 sets are very close and slight differences are

mainly explained by some differences in the number of assimilated observations. For REANA2, using a different background error-covariance matrix has direct consequences on these statistics. For radiosounding in the troposphere, AN departures are smaller for humidity (in figure 7, first row) but higher for wind (in figure 8, first row) due to the variations of the background error standard deviation described in section 2.2 : an increase for specific humidity and temperature (the background is less trusted and the resulting analysis is closer to observations), a decrease for vorticity and divergence directly related to the wind

field (the background is more trusted and the resulting analysis is farther from the observations). In both cases, this has a positive effect : for these two fields the subsequent 3-hour forecasts are closer to the observations as indicated by lower FG departures, even for the wind while the RMS of analysis-observations are higher. This results is enhanced by the use of high resolution vertical radiosondes which enable an increase of the observation number and a better comparison to the background than the TEMP message. For specific humidity, the RMS of AN and FG departure are respectively reduced by 30% and

15%. For wind, the differences are smaller and reach +20% for AN departure and -10% for FG departure. The impact of the background statistic changes is also visible for wind measurements from aircraft (Figure 8 second row) and radial velocity from Doppler radars (Figure 9), but less obvious for radar reflectivities (Figure 7, second row). The use of background error statistics more representative of the studied period allows for a better use of the observations.

Statistics on AN and FG departure are also informative on the quality of the additional observations only assimilated in

REANA2. For the second re-analysis, numerous wind profilers have been reprocessed and their number increased (Figure 8 second row). This better quality induces a decrease of FG departures and a reduction of AN departures, despite a higher background error for wind. Concerning the lidars (Figure 7 third row), it is worthy to note that the RMS background departures for BASIL and Leandre are very similar to the values obtained with radiosondes. WALI exhibits larger differences whose explanation is certainly linked to the fact that the lidar was located over land near the coast of the Menorca Island. Hence,







**Figure 7.** First guess (FG, solid lines) and analysis (AN, dashed lines) departure against radiosounding (mixing ratio (g/kg)) - row 1, against humidity derived from Doppler radar (humidity (percent)) - row 2, and against Lidars and dropsondes (mixing ratio (g/kg), only for REANA2) - row 3; columns correspond to mean departure (left), Root Mean Square departure (middle) and observations numbers (right). Black curves for SOP1, red for REANA1, blue for REANA2. Computation period extends from 05 September 2012 to 05 November 2012.



**Figure 8.** Statistics of zonal wind departures for SOP1 (black lines), REANA1 (red lines) and REANA2 (blue lines) for radiosondes (first row), aircraft (second row) and wind profiler (third row). Solid lines corresponds to first guess (FG) statistics and dashed lines to analyses statistics (AN).





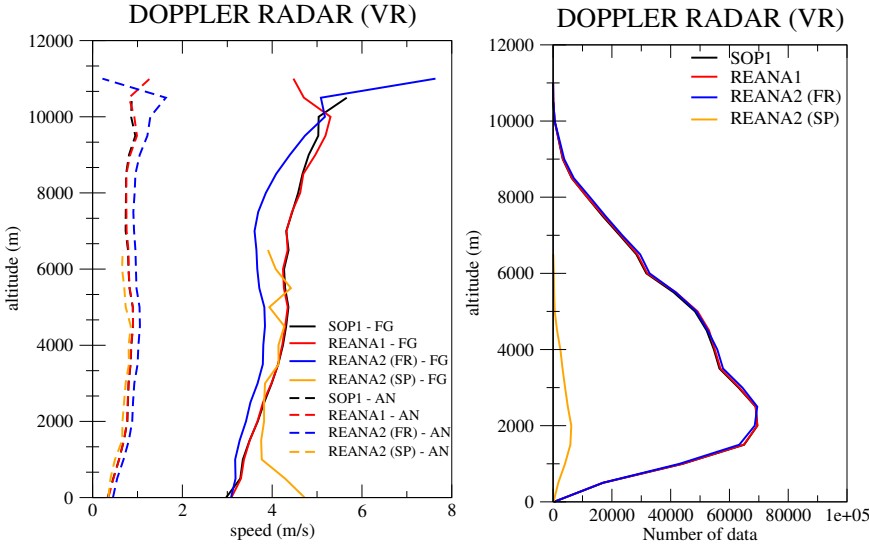

**Figure 9.** Root mean square departure for the Doppler wind between observations and background (solid line) and analysis (dashed lines) for SOP1 (black), REANA1 (red) and REANA2 (blue) over French radars and REANA2 over Spanish radars (orange) and observation number available in each data set (right panel).

the nearest AROME-WMED grid point is located over the Mediterranean Sea, which may introduce a discrepancy in the computation of the model equivalent, especially in the atmosphere low levels (boundary layer). Dropsondes have larger RMS differences (more than 2 g/kg between 800 and 1000hPa) than others radiosoundings (1.5 g/kg). This might be explained by the dropsonde sampling strategy, with launches close to convective areas, sampling low predictability areas, and leading to larger

humidity differences between the model and the observations. One can note that the AN departures are not impacted by these differences in the FG departure. Lastly, statistics for Spanish radar observations are compared to those of the French network (in figure 7 row 2 and figure 8 third row). Radar observations over Spain were available below 6000 m as a consequence of the sampling strategy. It appears that Spanish radar FG departure are higher than for French radars for Doppler wind below 2000 m and for reflectivities. Particularly, the latter ones exhibit a stronger dry bias (i.e. observation - background > 0) which

could be explained by a different observation preprocessing (in order to take into account the radar signal attenuation due to precipitation for example) for Spanish radars. If AN departures are increased for reflectivities, they remain very close to the French radar ones for radial velocity.

## 4.2 Surface parameter analysis and forecast

The surface observations used for the evaluation were extracted from the HyMeX database, which gathers the surface synoptic

observations over the HyMeX area, additional hourly observations of temperature and humidity at 2 m from Météo-France, AEMET and MeteoCat and the 10 m wind from some surface stations. The area selected for the evaluation is similar to the



HyMeX domain, i. e. 36 N-47.5 N, 9 W-17 E. The various forecasts were compared with observations up to the 54-h forecast range (REANA1 and REANA2), except for SOP1 which was only run up to the 48-h forecast range (Fig. 10).

The 2-m temperature exhibits a diurnal bias (forecast minus observations, Figure 10, model too cold at daytime and too warm at night-time) with a maximum absolute value of 0.5 $^o$C. REANA2 simulation has a noticeable reduced bias for each forecast range, which is a positive impact due to the modifications in the orography in REANA2. The standard deviation of forecast error, which increases with the forecast range, is also slightly reduced up to the 18-h forecast range. A bias reduction is also noticed for the 2-m relative humidity, together with a very small gain on the standard deviation (up to the 9-h forecast range). On the contrary, no real difference is noticeable for the biases between the three systems for the 10 m wind statistics. The relative improvement in forecast RMS error brought by REANA2 is larger than REANA1 one (more than 3% for temperature and humidity at the 3-h forecast range and 1% for the wind). The benefit varies as a function of the forecast range and remains up to the 30-h forecast range.

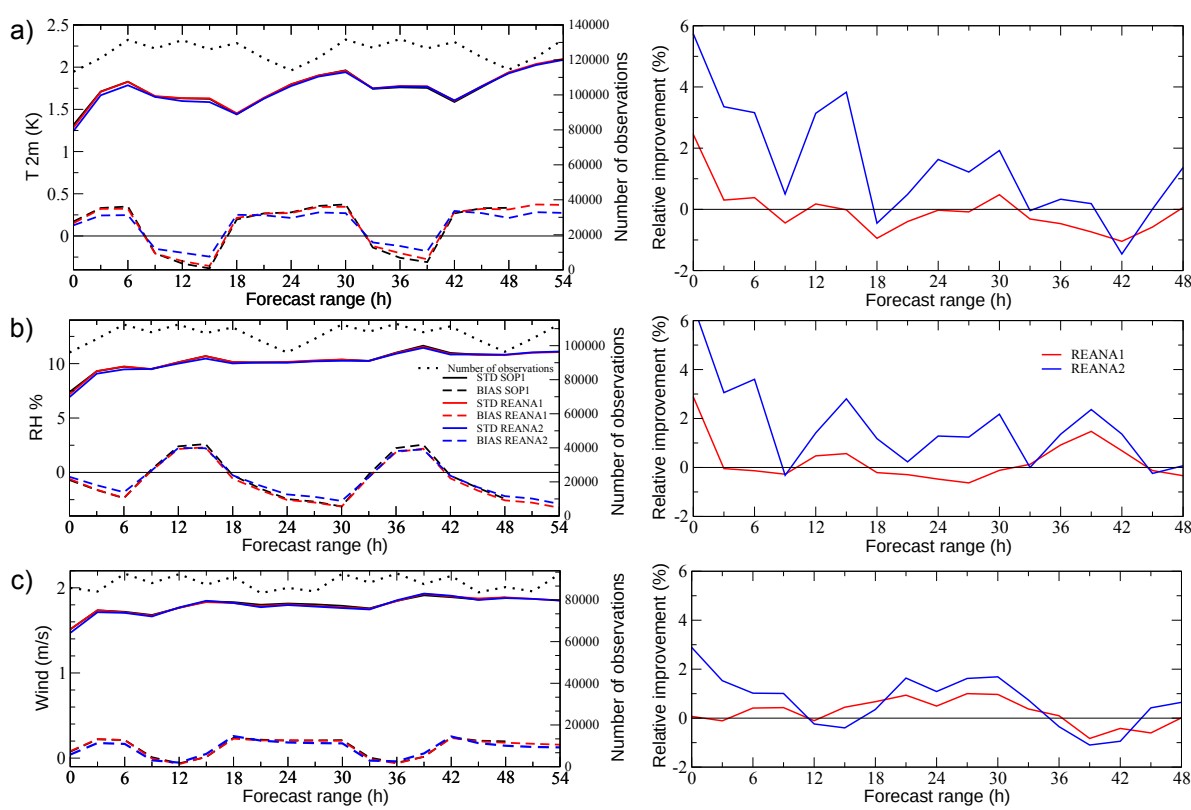

**Figure 10.** First column : bias (forecast - observation, dashed lines) and root mean square error (solid lines) computed for 2-m temperature (a), 2-m relative humidity (b) and 10-m wind speed (c) with respect to the forecast range for the real-time AROME-WMED model (SOP1, black), the first re-analysis (REANA1, red) and the second re-analysis (REANA2, blue) from 05 September to 05 November 2012. Dotted lines represent the number of observations used for the comparison (right vertical axis). Second column corresponds to the relative root mean square error difference calculated with respect to SOP1.



## 4.3 Upper level atmosphere/troposphere forecast

**Figure 11.** RMS forecast error computed with respect to radiosondes for the 24-h forecast range (first row), the 36-h forecast range (middle raw) and the 48-h forecast range (third row). First column represents temperature, middle one relative humidity and wind is plotted in the third column. Scores were computed from 5 September 2012 to 5 November 2012 and plotted in black for SOP1, in red for REANA1 and in blue for REANA2 from forecasts starting at 00UTC.





The forecast quality of the various AROME-WMED versions is fisrt assessed against radiosonde observations. Figure 11 gathers the RMS differences between AROME-WMED forecasts and radiosondes for temperature, relative humidity and wind at 24-h, 36-h and 48-h ranges. Overall, re-analyses forecast scores are improved compared to SOP1 ones. REANA1 improves the temperature forecast above 500 hPa at 24-h, the wind is improved over the whole troposphere but the maximum of im-
5 provement is found to be above 700 hPa, the gain brought by this re-analysis is significant according to a Bootstrap test at a 95% confidence level between 500 and 250 hPa. The improvement at 400 hPa is also significant. At 36-h, the gain is visible and is significant mostly for REANA2 below 600 hPa for temperature, at 800 and 300 hPa for the relative humidity and at 950 and 500 hPa for the wind. At 48-h REANA1 brought a significant improvement at 400 hPa for temperature and at 950, 500 and 250 hPa for the wind. REANA2 brought only an improvement at 100 and 600 hPa for the temperature.

10 ## 4.4 IWV

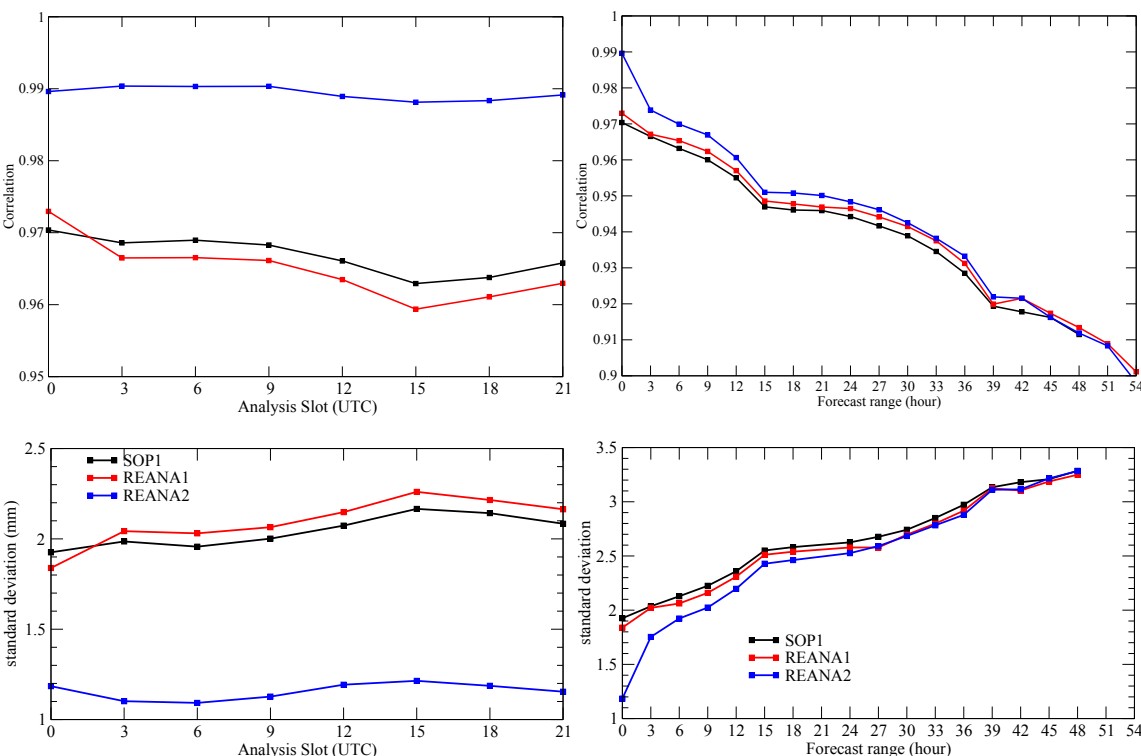

**Figure 12.** Correlation (upper panels) and standard deviations (bottom panels) of total integrated water vapour between AROME-WMED analyses (left panels) or forecasts (right panels) and GNSS observations (Bock et al. (2016)).

AROME-WMED model was also assessed using integrated water vapour (IWV) obtained from Version 1 data of GNSS ground based stations. IWV was indeed found to be linked with heavy precipitation, a maximum being observed before heavy precipitation event and a drop of its value occuring during the maximum of precipitation Bock et al. (2016). Results are





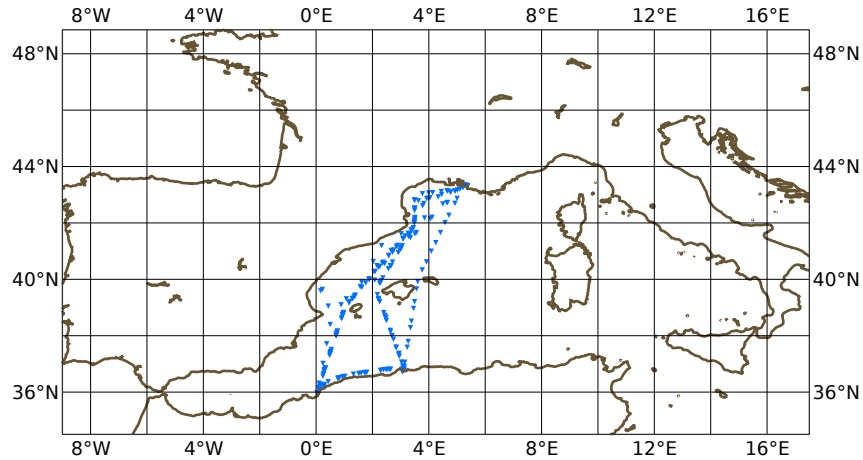

**Figure 13.** Locations of the Marfret-Niolon ZTD GNSS observations used for the comparison with AROME-WMED (re)-analyses during the period from 9 September 2012 00UTC to 1 November 2012 21 UTC.

presented in Fig. 12. These data being assimilated in REANA2 (and not in SOP1 and REANA1) the highest correlation (0.99) is found for each slot of the eight times of the REANA2 analysis. More than 32000 colocations were available to perform these computations. As expected, REANA1 and SOP1 correlations are lower(around 0.97); the maximum is observed for the 00UTC analysis slot and the minimum is noticed in the afternoon at 15UTC. The standard deviation of differences between

5   IWV analyses and observation is lower (between 1.1 and 1.2 mm) for REANA2 than for SOP1 and REANA1 (above 1.8 mm). The standard deviation is maximum at the 15UTC analysis slot (above 2 mm).

Concerning the forecast quality, as expected the IWV correlation between forecasts and observations decreases as the forecast range increases (from 0.99 down to 0.9 at 54-h). The largest score decrease is noticed in the very short forecast ranges. A diurnal cycle of the score is also found (local minima at +15 hour and +39 hour ranges); REANA1 is characterized by a

10   slightly higher correlation than SOP1 and the gain of REANA2 against REANA1/SOP1 is noticeable up to 24-h. The same conclusions apply for the standard deviation.

These results are confirmed over the sea with the validation against GNSS ZTD data (Figure 14), derived from a GNSS sensor on-board the Marfret-Niolon ship (Fig. 13). These data, which were not assimilated, represent an interesting independent source of validation. This data set is made of 418 measurements collected during the period from 9 September 2012 00UTC to

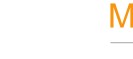
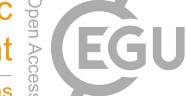


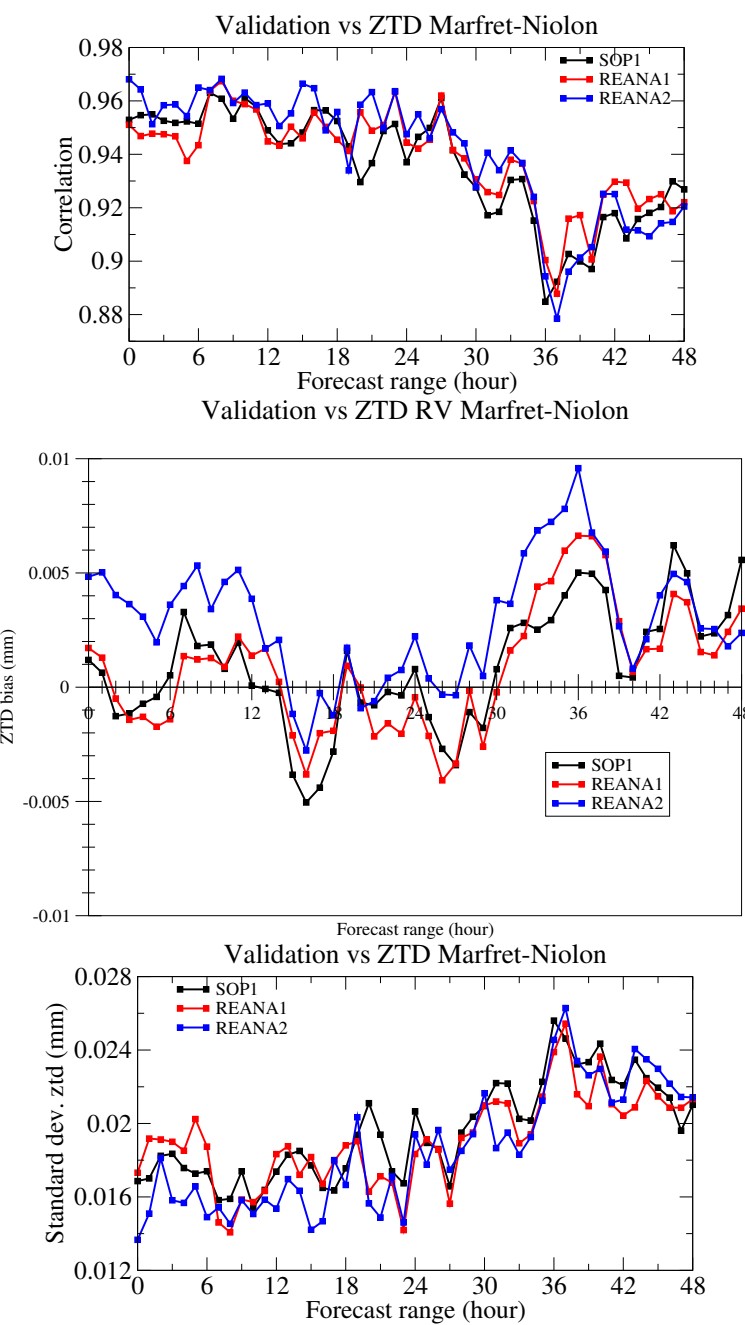

**Figure 14.** Verification with respect to GNSS zenithal total delay data from Marfret-Niolon ship as a function of the forecast range. Statistics of differences between re-analysis forecasts and observations are displayed in terms of correlation (top panel), mean (middle panel) and standard deviations (bottom panel) computed with all data available during the HyMeX 2-month period.





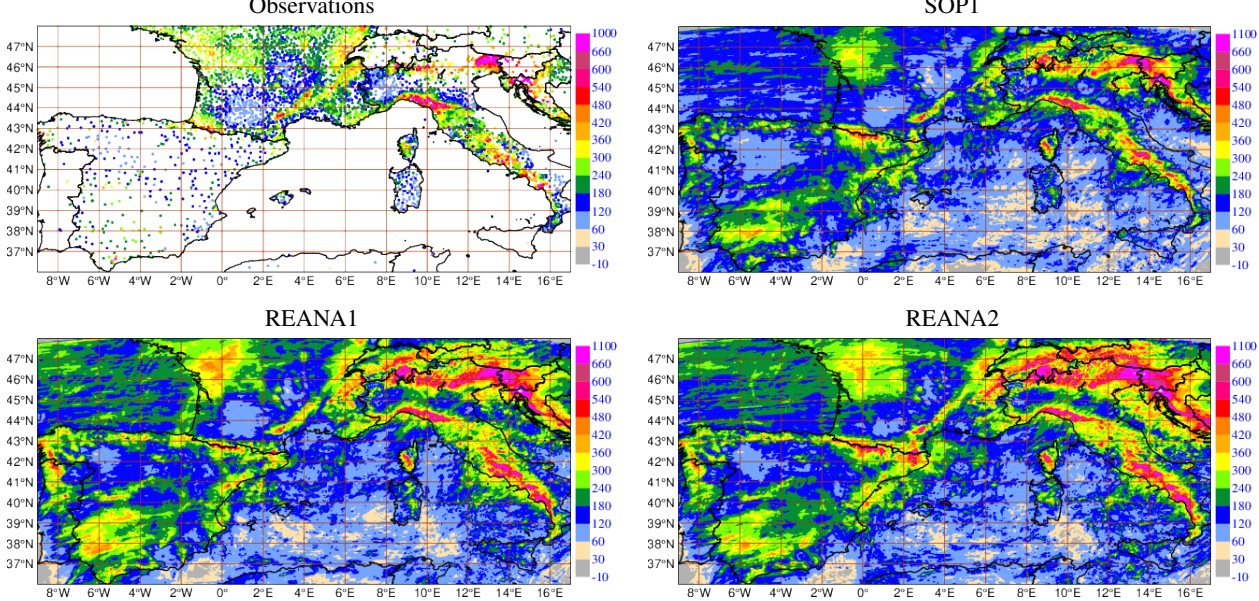

**Figure 15.** Precipitation amounts (mm) over a 2-month period from 5 September 2012 06 UTC to 5 November 2012 06 UTC, measured by surface stations. Accumulation between 06:00 and 06:00 next day (top left) predicted by SOP1 (top right), REANA1 (bottom left ) and REANA2 (bottom right).

1 November 2012 at 21 UTC and mainly in the western Mediterranean part of the AROME-WMED domain. Due to the small amount of data available, results are noisy. Nevertheless, it is noteworthy that the correlation between forecasts and observations is higher till the 24-h forecast range; standard deviation is lower up to the 24-h forecast range for REANA2 compared to SOP1 and REANA1. For the three simulations, a diurnal cycle of the ZTD bias exists. A stronger positive (moist) bias can be seen for the early forecast ranges of REANA2. At longer ranges the bias is more or less similar in the three simulations.

## 4.5 Surface precipitation

The evaluation is carried out with the 24-h accumulated precipitation (from 05 September to 05 November 2012) from the HyMeX database available in July 2017 (version4). These data were checked before computing scores. Only surface stations with daily precipitation for the full period (i. e. with an uninterrupted series) were taken into account. A good coverage is obtained over France, Italy and Spain (Fig. 15). REANA2 seems to yield more precipitation compared to the other versions, especially over elevated terrain. Even though the general precipitation pattern is similar in the three versions some differences can be noticed. For example, the maximum precipitation over Sardinia is not located at the same place. In REANA2 this local maximum is located in the northern part of the island, whereas in REANA1 it is located over the Eastern part. In addition, more precipitation are simulated over the sea in the Gulf of Lion for REANA2. The 2 month-period cumulated rainfall amount





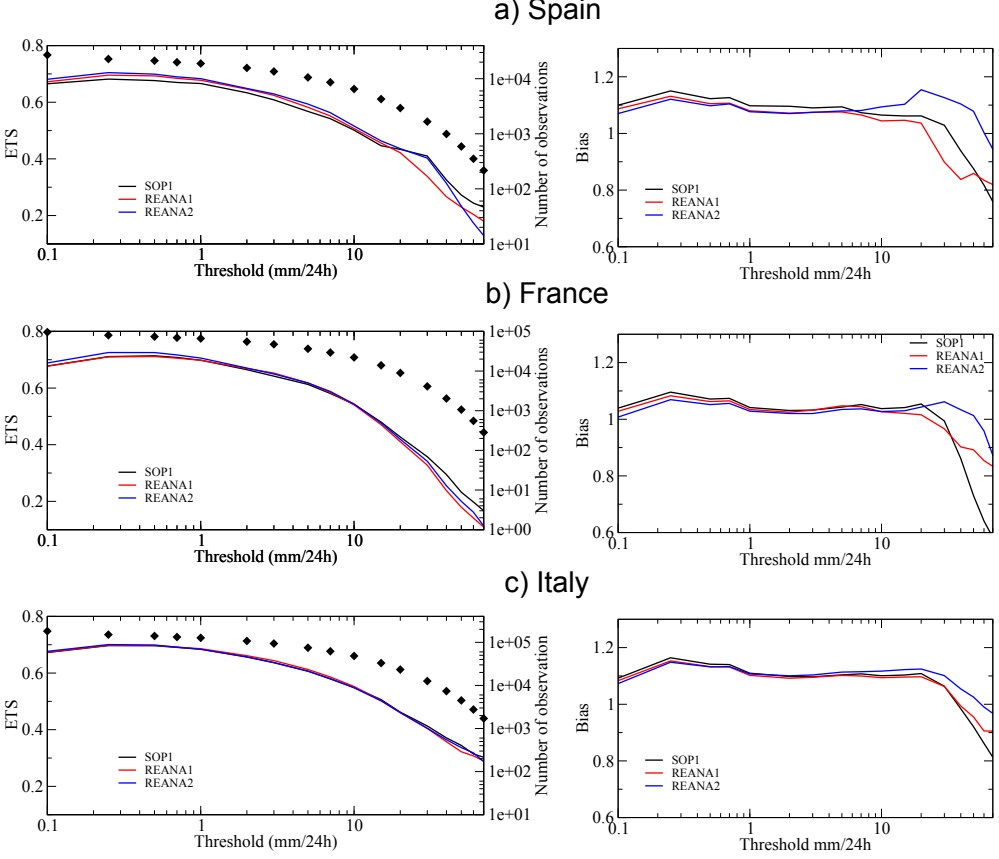

**Figure 16.** Equitable Threat Score (left panels) and bias (right panels) of the 06-30h accumulated precipitation simulated by AROME-WMED real-time version, REANA1 and REANA2 computed over Spain (panel a), France (panel b) and Italy (panel c) with rain gauges for the 2-month period during the HyMex campaign. Logarithm scale on $x$ axis. Diamonds represent the number of observations used for the comparison (logarithm scale).

shows some moister bias for REANA2 compared to REANA1 and SOP1 mainly over elevated terrain (Pyrénées, Alps, Sierra Nevada in Spain); some negative difference are found over Central italy and elsewhere (figure not shown).

Figure 16 shows the Equitable Threat Scores (ETS, definition given in appendix of Ebert (2008)) and the frequency bias for the 24-h accumulated precipitation computed with all data available in version4 for Spain, France and Italy. The closer to 1 the

5 ETS is, the better the forecast. Over Spain, the ETS is improved for both re-analyses and the gain is seen up to the 20 mm/24h threshold. The ETS for small thresholds are improved with REANA2 (up to 1mm/24h) over France but no improvement is seen over Italy. In the re-analyses, the frequency bias decreases up to the 5 mm/24h threshold over Spain and France and only for small thresholds (less than 1mm/24h) over Italy. For large thresholds, the frequency bias is larger for REANA2 than for the other two AROME-WMED versions. These results are in agreement with the overall accumulation of precipitation found in

10 Figure 15.



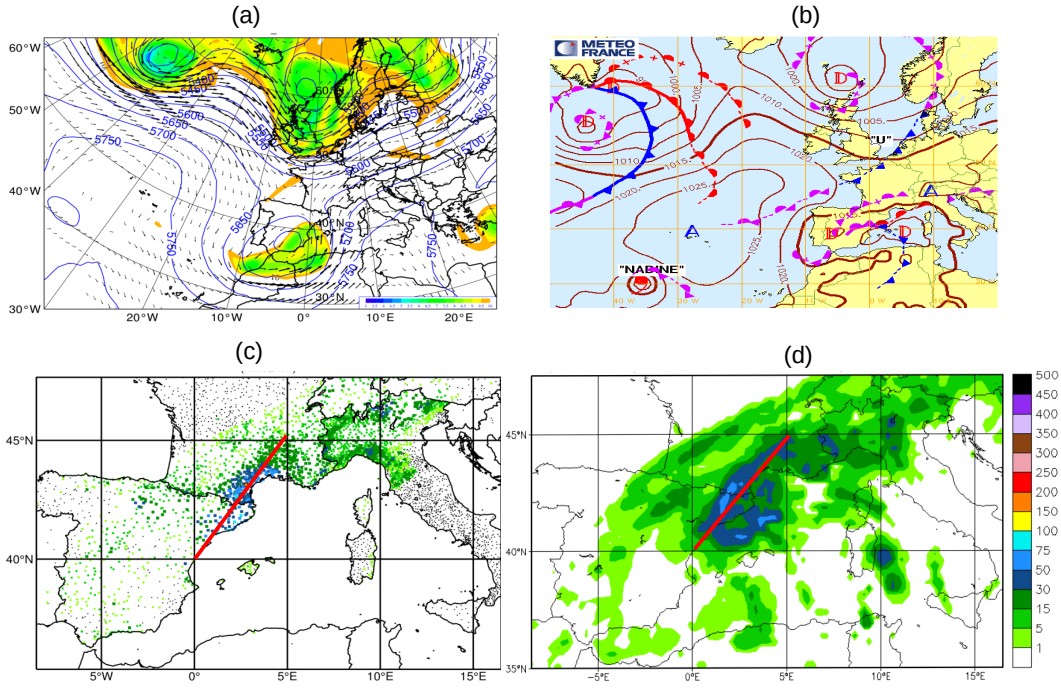

**Figure 17.** (a) 300 hPa wind (arrow), 1.5PVU surface height (colour) and 500hPa geopotential height (solid lines) and (b) surface synoptic conditions for 29 September 2012 00UTC. 29 September (c) daily observed precipitation and (d) 3B42 TRMM daily precipitation estimate.

## 5 IOP8 Qualitative evaluation

As illustrated in the quantitative forecast evaluation section, tiny improvements are noticed for REANA2 with respect to previous simulations for Quantitative Precipitation Forecasts (QPF). Such improvements in REANA2 can be found for specific periods of the HyMeX campaign. This is the case for IOP8, which took place during two days, from 28 to 29 September 2012. The key pattern of this IOP is a cut-off low centred to the south west of the Iberian Peninsula (28 September 00UTC) moving north-east and located over the Alboran Channel (29 September 00UTC). A detailed description of the early stages of IOP8 synoptic meteorological environment can be found in Bouin et al. (2017). Figure 17 depicts the large-scale synoptic conditions on 29 September 2012, 00UTC.

At low levels, on 29 September 00UTC, a weak complex surface low was positioned over the Gulf of Lion, in connection with the cut-off low as analysed by the global scale model ARPEGE. This cut-off drives a moist south easterly flow on its northeastern flank, towards the French Mediterranean coast, reinforced by orography (Cevennes ridge). On 29 September, this pressure minimum triggered heavy rainfall with embedded convection over the Gulf ol Lion (morning) and later on over the northern part of Catalonia and western part of Cevennes Vivarais. Daily precipitation amounts reaching 100 mm/24hr have been



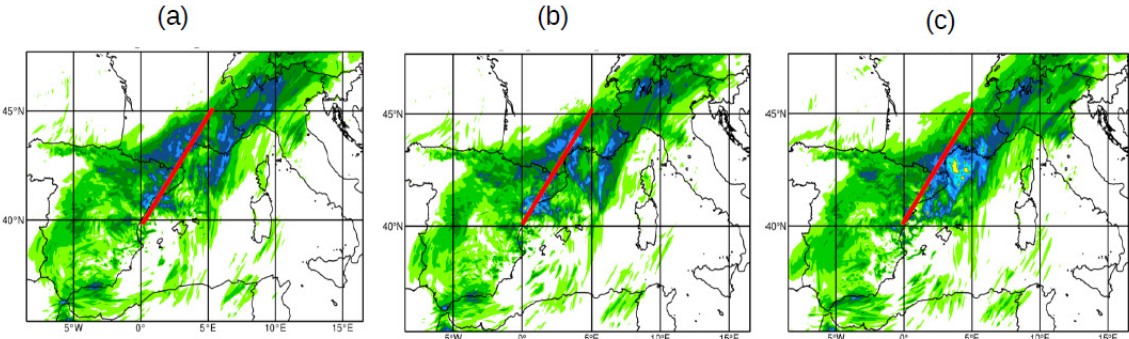

**Figure 18.** Daily precipitation amounts for 29 September 2012 simulated by the three different AROME-WMED versions: (a) SOP1, (b) REANA1 and (c) REANA2

recorded on the coastal zones along an axis from northern Catalonia to the Cevennes area, depicted by the red line extending from 40 N-0 E to 45 N-5 E in Fig. 17c. Such an amount of rainfall was also observed on the north-eastern part of the Gulf of Lion from the 3B42 TRMM estimates (Fig. 17d), whose estimates compare well, qualitatively and quantitatively, with in-situ measurements over land.

The daily accumulated precipitation amounts for the real time and first re-analysis exceeding 50 mm/day are shifted too far westward when compared to raingauges (Fig. 18 a and b). The maximum rainfall amount, located over the Gulf of Lion is better localized, though overestimated, in the second re-analysis (Fig. 18c). The ETS was computed for the various forecasts (00-24, and 24-48 hour range) valid for 29 September (00-24UTC period). The score was also computed for the 06-30 hour forecast range (corresponding to the 24-hour period between 29 September 06UTC to 30 September 06UTC). Figure 19 presents these

ETS curves; one can see that generally the re-analyses (1 or 2) perform better than the real-time version of AROME-WMED; surprisingly the ETS scores are better for the 24-48-hr forecast range than for the shorter (00-24-h) lead forecast period.

    The positive impact in QPF, may be linked to the better simulation of the deepening of the surface pressure low in the second re-analysis for the morning of 29 September located in the Gulf of Lion. At the Lion buoy (42.102N 4.703E), the minimum surface pressure observed on September 29th is 1008 hPa at 14UTC; the minimum surface pressure predicted by the

00UTC forecast initialized with the first re-analysis is 1010 hPa at 03UTC (not deep enough and too early), while the forecast simulation initialized at 00UTC by the second re-analysis indicates a minimum of 1008 hPa at 09UTC.

## 6   Conclusions

The AROME-WMED model was initially developed to study and forecast heavy precipitating events over the western Mediterranean basin. This model ran in real-time during both SOPs of HyMeX in Autumn 2012 and Winter 2013. Two re-analyses

were run after the HyMeX Autumn campaign. The first one was carried out just after the campaign to provide a same model configuration over the whole period, because an upgrade of the AROME-WMED version occurred during the period. In addi-





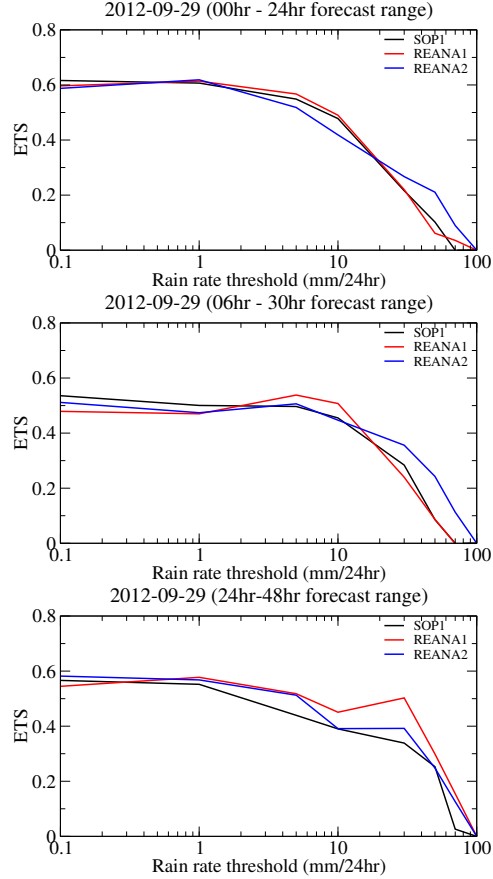

**Figure 19.** ETS scores valid for 29 September 2012 for simulated precipitation accumulated between 00-24-h (upper panel), 06-30-h (mid panel) and 24-48-h (lower panel) forecast ranges; initial conditions 29 September 2012 00UTC for upper and mid panels, 28 september 2012 00UTC for lower panel

tion a second re-analysis took into account as much data as possible from the experimental campaign (i. e. lidar and dropsonde humidity profiles) or from reprocessed data sets (such as GNSS ground stations ZTD, wind profilers, high resolution radioson-des, Spanish doppler radars). It also benefited from a more recent version of the AROME code, and from improved background error statistics computed over a 15-day period of the first HyMeX observing period.

5    The characteristics and the quality of the three AROME-WMED versions are discussed in this paper. More observations are assimilated in both re-analyses. The first re-analysis included 9% additional data, and the second re-analysis assimilated 24% more data. The use of background error statistics more representative of the studied period allows for a better use of the observations in the second re-analysis. The root mean square differences between first-guess simulations and observations are the smallest for the second re-analysis. The quality of research data such as lidar data is found to be comparable with the

10   operational radiosonde one.





The surface fields forecast is better for the second re-analysis; the 2m temperature diurnal bias is reduced. The forecast error standard deviation is improved for the first 18-h forecast range. Upper-level forecasts were compared to radiosondes observations and the forecast root mean square errors are decreased in the mid- and upper-troposphere for both re-analyses. The comparison with the reprocessed version 3 of GNSS data (Bock et al., 2016) shows that the second re-analysis IWV, in

terms of analyses and forecasts, is better correlated than the first one and the real-time version up to the 24-h forecast range. The standard deviation of IWV differences is also lower. Moreover, a comparison to GNSS zenithal total delay independent data (i.e. not assimilated) from vessel Marfret-Niolon also shows this positive impact up to +24hour. This is an interesting result over a sensitive area, where no conventional measurement are available. Larger values of accumulated precipitation during the 2-months period were obtained with the second re-analysis. Concerning the 24-hour precipitation evaluation, this positive impact

is less noticeable, but at least some improvement is diagnosed for the Iberian Peninsula and France for thresholds lower than 10 mm/24-h. The gain brought by the second re-analysis is smaller over Italy. Finally, the positive impact of second AROME-WMED re-analyse was detailed for the IOP8 high precipitating event which occurred over Spain and southern France, end of September 2012.

Studies are currently carried out to examine the respective impact of the additional observations such as reprocessed GNSS

data, high resolution radiosondes, radars and lidars assimilated in the second re-analysis.

*Data availability.* The source code of AROME-WMED being derived from the operational AROME one, cannot be obtained but the analyses and forecast fields are available in the HyMeX database (http:mistrals.sedoo.frHyMex) SOP1 doi:10.6096HYMEX.AROME_WMED.2012.02.20, REANA1 doi:10.6096HYMEX.REANALYSIS_AROME_WMED_V1.2014.02.10, REANA2: 10.14768MISTRALS-HYMEX.1492

*Competing interests.* No competing interests are present.

*Acknowledgements.* The authors would like to acknowledged the MISTRALS/HyMeX programme and the funding by ANR under contract IODA-MED ANR-11-BS56-0005 and MUSIC ANR-14-CE01-0014. Véronique Ducrocq, Jean-Francois Mahfouf and Jean-Antoine Maziejewski are warmly thanked for helping to improve the manuscript.



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
