# Peer review of "The AROME-WMED re-analyses of the first Special Observation Period of the Hydrological cycle in the Mediterranean experiment."

_Geoscientific Model Development, 2018_

## Referee Comment (RC1) · Anonymous Referee #1 · 28 Feb 2019

Review of the manuscript The AROME-WMED re-analyses of the first Special Observation Period of the Hydrological cycle in the Mediterranean experiment, by N. Fourrier et al.

This manuscript presents the results of the second Reanalysis performed with AROME-WMED model for the two-month period of the first Hymex field campaign SOP1. The results are contrasted with those available during the real time activity and those obtained with the first reanalyis. In particular, with respect to the latter, the new reanalysis employs a new B matrix for the 3DVar and assimilates a larger number of observations. Although the results show that the improvement of the second reanalysis is limited (I

would say less than expected, this is my feeling reading the paper), the methodology and the statistical results are suitably presented. I would suggest to highlight better what is really the valuable result of Reana2, since the effort was considerable. Moreover, some clarifications are required, as detailed in the following, together with minor corrections.

Abstract Line 15: remove upper-level

Introduction page 2 line 6: to study heavy precipitation line 24: provide acronyms of projects line 31: undertaken to exploit observations

Section 2 page 3 line 20: occur over the north western Mediterranean, from Catalonia... Otherwise it seems you are interested only on the event over the coast and I don't think it is true. page 4 Is the difference between the two orography computed on the raw (original) data, or on those interpolated on AROME grid? line 13: are temperature, specific humidity, the two horizontal component of the wind and surface pressure page 7 lines 8-11: I would remove the word resp. Brackets are clear enough.

Section 3 page 8 line 2: In addition, new line 3: between the two re-analyses line 13: radiosondes (available in France ...) were line 18: were also used in the second line 18: profilers data were carefully checked line 20: lidars were processed page 9 line 1: These data were smoothed. Moreover what does it mean? You mean they were interpolated at that resolution or did you used filter? lines 9-10: The higher amount of observations mainly comes .... profilers, satellite radiances, scatterometer wind estimates, Caption of Tab 2: between the first (REANA1) and the second (REANA2) reanalysis. The same for Fig. 5 caption.

Section 4 page 11 line 3: the performance of the data assimilation systems line 6: on figure 7 for observations related to humidity and on figures 8 and 9 for wind line 15: by the different number of assimilated observations, For REANA2, the use of a different lines 26-28: Does it mean that the difference between REANA2 and the others, with the same number of observations, indicate the impact of background statistics?

Please, make this point more clear. However, I can not see relevant differences in figs. 7 and 8, so maybe it would be better to stress where exactly you consider these differences relevant. line 30: the increase in wind profiler observations numbers is hardly visible in Fig. 8. Maybe it would be better to indicate explicitly the approximate number of observations. page 12: Caption of fig.7: explanation of the yellow lines in the middle panels is missing Fig. 7: the first impression looking at this picture is that the only relevant differences in the reanalyses are obtained when lidars are assimilated. I understand this is not the correct explanation of Fig. 7, but you should stress better the lidar impact in the text. page 14: lines 1-4: I can not understand if the impact of dropsondes is retained and considered positive or not. lines 7: the reference to fig.8 third row is correct for radar? page 15 line 3: more than a diurnal bias, I would say that the model underestimates the diurnal cycle line 5: how can you ascribe the impact to the different orography. Is there any clue? line 6: slightly reduced. This is hardly visible in the plot. Is this significant? lines 10-11: not true for wind page 17 line 3: re-analyses forecast: I believe forecasts starting from re-analyses is better. lines 7-9: please revise this part. I can not find correspondence with the figures. For example for temperature I would say above 800 hPa instead of below 600 hPa as you wrote last line: (Bock et al., 2016) page 18 line 1: These data being assimilated ... Are you assimilating GPS in REANA2 and then comparing results against the same data? Is this fair? page 20 line 13: eastern part page 21 line 1: some moister bias: do you see this bias from Fig.15? In case, I believe it very difficult to make a fair comparison between the three reanalyses and observations, given missing data over the sea and over large portion of the land area. A statistical quantitative comparison would be needed to draw conclusions. line 2: Central Italy

Section 5 page 22 line 5: IOP was line 9: Lion, associated with line 10: cut-off drove line 11: how can be the SEly flow reinforced by the orography of Cevennes? line 13: 100 mm/24h were recorded page 23 line 3: that compare well line 11: remove lead lines 10-11: results are worse for low precipitation values in the 6-30h range. I was wondering how different is the number of raingauges in 0-24 and 6-30 periods. Does

it have an impact on the statistics? lines 12-16: The explanation of positive impact on QPF should be better supported. Here it is just an hypothesis and no one can say if it's true or not. Please provide more details and more evidence.

Conclusions page 25 line 8: Larger values: are these values too large or not? It is not clear.

---

## Referee Comment (RC2) · Anonymous Referee #2 · 15 Apr 2019

This paper aims at assessing the main differences between the real-time version of the AROME-WMED model and two distinct re-analyses strategies during the first special observation period (SOP1) that took place in Autumn 2012 (05 September to 06 November). In particular, it is highlighted that the second re-analyses assimilates much more observations than the first re-analyses (in particular, it uses 15% of additional information), and it also uses an updated version of the real-time version of the AROME-WMED model. Furthermore, the second re-analyses also uses a more realistic background error covariance matrix, that plays a crucial role in the data assimilation process, and allows that a higher number of observations can be assimilated. As it is expected, due to the usage of a more realistic background error covariance matrix and

the assimilation of a higher number of observations from different instruments, results obtained from the second re-analyses depicts the best verification scores.

This study is interesting but additional work is needed to improve the quality of the present manuscript to be considered in this journal. Main concerns and some suggestions are listed below. Taking into consideration these comments, I recommend some minor modifications before it can be accepted for publication in the Geoscientific Model Development journal.

General comments:

As it is state above, this study clearly shows the benefits of the second re-analysis in comparison with the first re-analysis and the real-time version of the AROME-WMED model. However, it is not explained which factors (i.e., topography, background error covariance matrix, type of observations assimilated or number of observations assimilated) have played a key role in the improvement of the second re-analysis. With the main objective of improving the quality of the manuscript, a more detailed discussion about the main reasons of these benefits should be addressed performing some numerical sensitivity experiments. For instance, if the second re-analysis used the same topography (GTOPO30) and assimilates the same type and number of observations than the first re-analysis, would the results be very different from the obtained originally? In this example, the differences obtained could be attributable to the effect of the background error covariance matrix.

Regarding the implementation of the 3DVar data assimilation technique, no information about the observational errors assigned to the different kind of data assimilated is provided along the entire manuscript. Taking into account the relevant role of this parameters in the effectiveness of the data assimilation algorithm, I strongly suggest the authors to add this information.

Minor comments:

The following are some suggestions that could help to improve the quality of the manuscript:

Introduction Section:

1) Page 3 (line 3): "the AROME-WMED re-analyses and the real-time versions The different..." –> "the AROME-WMED re-analyses and the real-time versions. The different..."

2) Page 3 (line 6): "Intensive Observation Period (IOP) 8" "Intensive Observation Period (IOP8)"

3) Page 3 (Table 1): Remove open parenthesis " ( " appeared in the REANA1 box. This open parenthesis should be located in the REANA2 box: "(from 17 to 31 October 2012)". Also, the caption is located very close to the table. Add some additional vertical space between them.

Description AROME-WMED Model Section:

4) Page 3 (line 21): "The model grid includes a 960x640 point matrix..." "The horizontal model grid includes a 960x640 point matrix..."

5) Page 4 (Figure 1): Add label to the left panel colorbar. In addition, add some extra horizontal white space between panels, they are quite close.

6) Page 4 (line 9): Add space after the second 06 UTC: "period 06 UTC-06UTC on the following day" "period 06 UTC-06 UTC on the following day"

7) Page 4 (line 12): It is stated that an assimilation window of +/- 1h30 is used. Is this assimilation window used indistinctly for all types of observations? Observations with high temporal resolution, such as radar observations, should not use this large assimilation window. Could the authors provide detailed information of how they apply this assimilation window?

8) Page 4 (line13): "analysed parameters are temperature,..." "analysed variables??

are temperature,. . ."

9) Page 5 (Figure 2): Add a), b), c) and d) labels to panels.

10) Page 6 (Figure 3): Same that in Figure 2.

11) Page 7 (line 13): "horizontal correlation length-scales are slightly longer". Do the authors refer to the horizontal correlation scales from REANA2? Please improve this sentence.

Assimilated Data Section:

12) Page 8 (Table 2): The caption is located very close to the table. Add some additional vertical space between them.

13) Page 9 (line 9): igher higher

Assimilation Results Section:

14) Page 20 (line 14): cumulated accumulated

15) Page 20 (Figure 15): Add labels to figure colorbars

IOP8 Qualitative Evaluation Section:

16) Section title: IOP8 Qualitative evaluation IOP8 qualitative evaluation

17) Page 22 (line 12): Gulf ol Lion Gulf of Lion

18) Page 23 (lines 10-11): Regarding ETS verification score obtained from the daily accumulated precipitation amounts exceeding 50 mm/day, it is stated that ETS scores are better for the 24-48 hour forecast range than for the 00-24 hour forecast period. Could the authors provide some explanation of this result?

Please also note the supplement to this comment:
https://www.geosci-model-dev-discuss.net/gmd-2018-303/gmd-2018-303-RC2-supplement.pdf

---

## Author Comment (AC1) · 3 Jun 2019

We thank Reviewer 1 for his/her comments which helped to improve, we hope, the quality of the manuscript. Reviewer 1's comments are in bold font, our answers are written with normal font.

**This manuscript presents the results of the second Reanalysis performed with AROMEWMED model for the two-month period of the first Hymex field campaign SOP1. The results are contrasted with those available during the real time activity and those obtained with the first reanalyis. In particular, with respect to the latter, the new reanalysis employs a new B matrix for the 3DVar and assimilates a larger number of observations.**
**Although the results show that the improvement of the second reanalysis is limited (I would say less than expected, this is my feeling reading the paper), the methodology and the statistical results are suitably presented. I would suggest to highlight better what is really the valuable result of Reana2, since the effort was considerable. Moreover, some clarifications are required, as detailed in the following, together with minor corrections.**
We have extended the conclusion of the paper to better highlight the results found in this paper :

« The AROME-WMED model was initially developed to study and forecast heavy precipitating events over the western Mediterranean basin in the frame of the HyMeX programme . This model ran in real-time during both SOPs of HyMeX in Autumn 2012 and Winter 2013. Two re-analyses were run after the HyMeX Autumn campaign. The first one was carried out just after the campaign to provide a same model configuration over the whole period, because an pgrade of the AROME-WMED version occurred during the period. In addition a second re-analysis was perfomed a few years after and took into account as much data as possible from the experimental campaign (i. e. lidar and dropsonde humidity profiles) or from reprocessed data sets (such as GNSS ground stations ZTD, wind profilers, high vertical resolution radiosondes, Spanish doppler radars). It also benefited from a more recent version of the AROME code including a orography change, and from improved background error statistics computed over a 15-day period of the first HyMeX observing period. The analysis and forecast fields of these three AROME-WMED versions are available in the HyMeX database (http://mistrals.sedoo.fr/HyMex)).
 The characteristics and the quality of the three AROME-WMED versions are discussed in this paper. More observations are assimilated in both re-analyses. The first re-analysis included 9% additional data, and the second re-analysis assimilated 24% more data. These data in the case of REANA2 mainly came from GNSS ground station, radiosondes and satellite radiances. The use of background error statistics, more representative of the studied period, allows a better use of the observations in the second re-analysis. The root mean square differences between first-guess simulations and observations are the smallest for the second re-analysis. Depending on the change of the background statistics, the root mean square differences between analysis simulations and observations are adjusted . The observation departure study showed that the research data quality such as lidar data is found to be comparable with the operational radiosonde one. Concerning the forecast quality, the surface field forecast is better for the second re-analysis; the 2m temperature diurnal bias is reduced up to the 54-h forecast range . The forecast error standard deviation is improved for the first 18-h forecast ranges. This improvement is mainly due to the orography change in REANA2. A reduction of the 2-m relative humidity bias is also found.
 Larger values of accumulated precipitation during the 2-months period were obtained with the second re-analysis and the comparison with observations suggest an overestimation of large precipitation amount mainly over relief. However the frequency bias is decreased for smaller thresholds, over the AROME-WMED domain . Concerning the 24-hour precipitation evaluation, this positive impact is less noticeable, but at least some improvement is diagnosed for the Iberian Peninsula and France for thresholds lower than 10 mm/24-h. The gain brought by the second re-analysis is smaller over Italy. Finally, the positive impact of second AROME-WMED re-analyse was detailed for the IOP8 high precipitating event which occurred over Spain and southern France, end of September 2012.
Preliminary studies with data assimilation experiments with only the version code change including the new backgrounds statistics have shown that the gain in forecast score brought by REANA2 is due to the new observations assimilated and the new code version. Figure 22 illustrates this fact for the 36-h forecast range. A small reduction of the root mean square error is obtained with the assimilation of new observations in temperature and for the wind in the troposphere. The improvement brought by the observations is less clear for the humidity. Concerning the 24-h accumulated precipitation, REANA2 improves small thresholds (0.5, 1 mm/24h) compared to the preliminary experiment, REANA1 and SOP1. It is clear that the 2-m temperature and humidity forecast bias improvement is related to the orography change. The improvement found in the REANA2 fields is therefore the result of all the changes made compared to REANA1 and SOP1. »

**Abstract Line 15: remove upper-level.** Done

**Introduction page 2 line 6: to study heavy precipitation** Changed
**line 24: provide acronyms of projects** Fronts and Atlantic Storm-Track EXperiment (FASTEX) and Mesoscale Alpine Programme (MAP)

**line 31: undertaken to exploit observations** Done

**Section 2 page 3 line 20: occur over the north western Mediterranean, from Catalonia… Otherwise it seems you are interested only on the event over the coast and I don't think it is true.** Thanks, it was modified.

**page 4 Is the difference between the two orography computed on the raw (original) data, or on those interpolated on AROME grid?**
The difference computed here are the differences computed between both orographies interpolated on the regular AROME-WMED grid. This specification was added in the text : « A mean difference of 21 m was found between the orography interpolated onto the AROMEWMED grid from GMTED2010 used in the REANA2 and the one interpolated from GTOPO30 used in the REANA2 and SOP1 versions.

**line 13: are temperature, specific humidity, the two horizontal component of the wind and surface pressure.** Done

**page 7 lines 8-11: I would remove the word resp. Brackets are clear enough.** Done

**Section 3 page 8 line 2: In addition, new**
**line 3: between the two re-analyses** Changes were done.

**line 13: radiosondes (available in France ...) were** Done
**line 18: were also used in the second** Modified
**line 18: profilers data were carefully checked** Changed
**line 20: lidars were processed** Done
**page 9 line 1: These data were smoothed.** Done. **Moreover what does it mean? You mean they were interpolated at that resolution or did you used filter?** Lidar data were smoothed through using a vertical interpolation at a 200m vertical resolution. This was specified in the text.

**lines 9-10: The higher amount of observations mainly comes .... profilers, satellite radiances, scatterometer wind estimates,** Suggested changes were done

**Caption of Tab 2: between the first (REANA1) and the second (REANA2) reanalysis. The same for Fig. 5 caption.** Done

**Section 4 page 11 line 3: the performance of the data assimilation systems** Done
**line 6: on figure 7 for observations related to humidity and on figures 8 and 9 for wind.** The change was made.

**Line 15: by the different number of assimilated observations, For REANA2, the use of a different.** Done

**lines 26-28: Does it mean that the difference between REANA2 and the others, with the same number of observations, indicate the impact of background statistics? Please, make this point more clear. However, I can not see relevant differences in figs. 7 and 8, so maybe it would be better to stress where exactly you consider these differences relevant.**
Yes the change are mainly brought by the change of background statistics. The paragraph was rewritten as follows : « The impact of the background statistic changes is also visible for wind measurements from aircraft (Figure 8 second row) whose number is similar between the three experiments and radial velocity from Doppler radars (Figure 9). The REANA2 AN departures are slightly larger than the SOP1 and REANA1, but the subsequent FG departures are smaller for the REANA2 than for REANA1 and SOP1 between 800 and 300 hPa. The reduction in humidity AN departures is less obvious for radar reflectivities (Figure 7, second row). These results suggest that the use of background error statistics more representative of the studied period allows for a better use of the observations. »

**line 30: the increase in wind profiler observations numbers is hardly visible in Fig. 8. Maybe it would be better to indicate explicitly the approximate number of observations.**
The text was rephrased : « For the second re-analysis, numerous wind profilers have been reprocessed and their number increased from 1,000 to 4,000 observations at 700 hPa (Figure 8 third row). The reference to

the row of the picture was wrong and refered to the assimilated aircraft data of which number is similar in the three experiment.

**page 12: Caption of fig.7: explanation of the yellow lines in the middle panels is missing**
The caption of fig. 7 was extended : **«** First guess (FG, solid lines) and analysis (AN, dashed lines) departure against radiosounding (mixing ratio (g/kg)) - row 1, against humidity derived from Doppler radar (humidity (percent)) - row 2, and against Lidars and dropsondes (mixing ratio (g/kg), only for REANA2) - row 3; columns correspond to mean departure (left), Root Mean Square departure (middle) and observations numbers (right). In the first two rows, black curves are for SOP1, red for REANA1, blue for REANA2. Orange lines are for Spanish radars in REANA2. »

**Fig. 7: the first impression looking at this picture is that the only relevant differences in the reanalyses are obtained when lidars are assimilated. I understand this is not the correct explanation of Fig. 7, but you should stress better the lidar impact in the text.**
The panels were retraced to highlight differences.  « Concerning the lidars (Figure 7 third row), it is worthy to note that the RMS background departures for BASIL and Leandre are very similar to the values obtained with radiosondes (Fig. 7 first raw) showing data of comparable quality . WALI exhibits  larger differences whose explanation is certainly linked to the fact that the lidar was located over land near the coast of the Menorca 10  Island. Hence, the nearest AROME-WMED grid point is located over the Mediterranean Sea, which may introduce a discrepancy in the computation of the model equivalent, especially in the atmosphere low levels (boundary layer). It should be also mentioned that lidar data represent very few data among the total number of assimilated data .

**page 14: lines 1-4: I can not understand if the impact of dropsondes is retained and considered positive or not.**
The paragraph was rerewritten to provide a clearer message : « Dropsondes exhibit a larger bias and RMS differences (more than 2 g/kg between 800 and 1000hPa) than radiosoundings (1.5 g/kg). Dropsonde measurements are therefore farer from the model values. This might be explained by the dropsonde sampling strategy, with launches close to convective areas, sampling low predictability areas, and leading to larger humidity differences between the model and the observations. However one can note that the AN departures are not impacted by these differences in the FG departure. »

**lines 7: the reference to fig.8 third row is correct for radar?** No the correct reference is Figure 9,first row. It was modified in the text.

**page 15 line 3: more than a diurnal bias, I would say that the model underestimates the diurnal cycle**
It was modified.
**line 5: how can you ascribe the impact to the different orography. Is there any clue?** A previous experiment was performed with modification including only the orography and the background matrix ; the impact on the 2-m temperature is already present in this sensitivity test.

**line 6: slightly reduced. This is hardly visible in the plot. Is this significant?**
Yes it is statistically significant according a bootstrap test. This was specified in the text : « The standard deviation of forecast error, which increases with the forecast range, is also slightly and reduced up to the 18-h forecast range and this is statistically significant according a Bootstrap test . »

**lines 10-11: not true for wind.** That is true and was added in the text ?

**page 17 line 3: re-analyses forecast: I believe forecasts starting from re-analyses is better.** We agree with the reviewer and have changed the formulation : scores of forecast starting from re-analyses are improved compared to those starting from SOP1

**lines 7-9: please revise this part. I can not find correspondence with the figures. For example for temperature I would say above 800 hPa instead of below 600 hPa as you wrote last line**
This part was revised as follows :  At 36-h, the improvement brought by REANA2 with respect to SOP1 and REANA1 is noticeable all along the troposphere for temperature, humidity and wind, except for temperature between 800 and 900 hPa and above 500 hPa for relative humidity where REANA1 provides improved forecast. In addition, REANA1 forecast is better than SOP1 but generally in a less extend than REANA2.

At 48-h range, REANA1 and REANA2 improve the temperature forecast above 700hPa, the humidity forecasts are not improved, but wind forecast is improved above 600 hPa. These results are statistically

significant ( 95% confidence Bootstrap test) for temperature at 100hPa . REANA2 brought only a significant improvementat 600 and 100 hPa and near the surface for temperature.

**page 18 line 1: These data being assimilated ... Are you assimilating GPS in REANA2 and then comparing results against the same data? Is this fair?** Yes we do in order to provide the best final reanalysis (REANA2). We agree that it is not fair for the two other experiments, but we can compare REANA1 and SOP1 together as these data represent independent data for both experiments. The data provided by the GPS sensor on board the Marfret Niolon ship were not assimilated and thus represent an independent source of data.

**page 20 line 13: eastern part** Modified

**page 21 line 1: some moister bias: do you see this bias from Fig.15?**

[Figure]

You are right it is not possible to see it from Fig 15. However we have the picture above which shows the differences .of accumulated precipitations. We propose to replace the sentence with. « The 2-month-period accumulated rainfall amount shows larger precipitation values for REANA2 than for REANA1 and SOP1, mainly over elevated terrain (Pyrénées, Alps, Sierra Nevada in Spain); some negative difference are found over Central Italy and elsewhere (figure not shown) and also over the Gulf of Lion.

**In case, I believe it very difficult to make a fair comparison between the three reanalyses and observations, given missing data over the sea and over large portion of the land area. A statistical quantitative comparison would be needed to draw conclusions.**

[Figure]

The figure above shows the frequency bias computed with respect to the raingauges over the AROME-WMED domain. This bias is improved for small thresholds (<1mm/24h) in the REANA2 and these results are statistically significant. The degradation for thresholds exceeding 1mm/24h in the REANA2 is not significant due to the lower number of observations.
The figure and its comment was added in the paper in section 4.5.

**line 2: Central Italy.** Modified.

**Section 5 page 22 line 5: IOP was** Modified.

**line 9: Lion, associated with.** Changed.

**line 10: cut-off drove.** Modified.

**line 11: how can be the SEly flow reinforced by the orography of Cevennes?** It is due to the barrier effect that produces the Cevennes mountains (Buzzi et al (2003). The reference was added in the text : « reinforced by orography (Cevennes ridge, which induced a barrier effect as shown in Buzzi et al. (2003))

**Line 13: 100 mm/24h were recorded** Modified.

**page 23 line 3: that compare well** Changed.

**line 11: remove lead** Done

**lines 10-11: results are worse for low precipitation values in the 6-30h range. I was wondering how different is the number of raingauges in 0-24 and 6-30 periods. Does it have an impact on the statistics?**
It is known that some uncertainty/variability exists in the statistics. For the 00-24 h range, there are around 4500 observations a day available, for model comparison ; for the 06-30h there are around 6000 observations available. We do not believe that this difference has a strong impact on the statistics.

**lines 12-16: The explanation of positive impact on QPF should be better supported. Here it is just an hypothesis and no one can say if it's true or not. Please provide more details and more evidence.**
Our explanation is a guess but we are not sure. A sensivity study would help to understand in depth the differences (beyong the score of this study).

**Conclusions page 25 line 8: Larger values: are these values too large or not? It is not clear.**
The study of the frequency bias suggest that the largest valued of accumulated observations are overestimated. The sentence was completed with : « and the comparison with observations suggest an overestimation of large precipitation amount over relief »

---

## Author Comment (AC2) · 3 Jun 2019

**Referee2**

We thank Reviewer 2 for his/her comments which helped to improve, we hope, the quality of the manuscript. Reviewer 2's comments are in bold font, our answers are written with normal font.

**This study is interesting but additional work is needed to improve the quality of the present manuscript to be considered in this journal. Main concerns and some suggestions are listed below. Taking into consideration these comments, I recommend some minor modifications before it can be accepted for publication in the Geoscientific Model Development journal.**

**General comments:**
**As it is state above, this study clearly shows the benefits of the second re-analysis in comparison with the first re-analysis and the real-time version of the AROME-WMED model. However, it is not explained which factors (i.e., topography, background error covariance matrix, type of observations assimilated or number of observations assimilated) have played a key role in the improvement of the second re-analysis. With the main objective of improving the quality of the manuscript, a more detailed discussion about the main reasons of these benefits should be addressed performing some numerical sensitivity experiments. For instance, if the second re-analysis used the same topography (GTOPO30) and assimilates the same type and number of observations than the first re-analysis, would the results be very different from the obtained originally? In this example, the differences obtained could be attributable to the effect of the background error covariance matrix.**

The second reanalysis was built in two steps. The first step has consisted in changing the AROME code version and the background error covariance statistics. Then many experiments were carried out in parallel to add the observations. It took a long time to run the experiments over a more or less long period.

The benefits of the second re-analysis come from many components. Preliminary studies with an experiment with no adddition of new observations have shown that the reduction in the bias of the temperature at 2 m comes from the new orograpĥy. The conclusion section has been extended with a figure (see below) and a discussion to highlight the results o REANA2.

[Figure]

Fig 22 36-h forecast Root Mean Square errors with respect to radiosondes for temperature, humidity and wind.

Preliminary studies with data assimilation experiments with only the version code change including the new backgrounds statistics have shown that the gain in forecast score brought by REANA2 is due to the new observations assimilated and the new code version. Figure 22 illustrates this fact for the 36-h forecast range. A small reduction of the root mean square error is obtained with the assimilation of new observations in temperature and for the wind in the troposphere. The improvement brought by the observations is less clear for the humidity. Concerning the 24-h accumulated precipitation, REANA2 improves small thresholds (0.5, 1 mm/24h) compared to the preliminary experiment, REANA1 and SOP1. It is clear that the 2-m temperature and humidity forecast bias improvement is related to the orography change. The improvement found in the REANA2 fields is therefore the result of all the changes made compared to REANA1 and SOP1. »

A second paper (companion paper) is underway to show the benefit brought by some observation data sets and a reference to this paper is made in the conclusion.

**Regarding the implementation of the 3DVar data assimilation technique, no information about the observational errors assigned to the different kind of data assimilated is provided along the entire manuscript. Taking into account the relevant role of this parameters in the effectiveness of the data assimilation algorithm, I strongly suggest the authors to add this information.**

We now provide  a figure of observation errors assigned to the different datasets and a comment in the paper.

[Figure]

The associated observation errors were deduced from the monitoring of standard deviation of differences between background simulations and observations and they are displayed in Figure 5. Some differences are observed on the plot for lidar data. The observation error for Leandre II data are smaller than the other ones and WALI assigned observation error is slightly larger than BASIL and TEMP ones. Concerning temperature and wind, the assigned observation errors are the same for dropsondes, radiosondes and profilers ; the aircraft data errors are larger.

**Minor comments:**
**The following are some suggestions that could help to improve the quality of the manuscript:**

**Introduction Section:**
**1) Page 3 (line 3): "the AROME-WMED re-analyses and the real-time versions The different. . ." –> "the AROME-WMED re-analyses and the real-time versions. The different. . ."** The point at the end of the sentence was added.
**2) Page 3 (line 6): "Intensive Observation Period (IOP) 8" "Intensive Observation Period (IOP8)"** The change was made
**3) Page 3 (Table 1): Remove open parenthesis " ( " appeared in the REANA1 box. This open parenthesis should be located in the REANA2 box: "(from 17 to 31 October 2012)". Also, the caption is located very close to the table. Add some additional vertical space between them.**
The modifications were made

**Description AROME-WMED Model Section:**
**4) Page 3 (line 21): "The model grid includes a 960x640 point matrix. . ." "The horizontal model grid includes a 960x640 point matrix. . ."** « Horizontal » was included between the and model.
**5) Page 4 (Figure 1): Add label to the left panel colorbar. In addition, add some extra horizontal white space between panels, they are quite close.** The suggested modification were done Figure 1.
**6) Page 4 (line 9): Add space after the second 06 UTC: "period 06 UTC-06UTC on the following day" "period 06 UTC-06 UTC on the following day"**Done
**7) Page 4 (line 12): It is stated that an assimilation window of +/- 1h30 is used. Is this assimilation window used indistinctly for all types of observations? Observations with high temporal resolution, such as radar observations, should not use this large assimilation window. Could the authors provide detailed information of how they apply this assimilation window?** The reviewer is right, this assimilation window is applied differently with respect to the observation type. For non frequent observations at the same location, all observations included in this time range are considered. However for frequent

observation types such as radars or radiances from geostationnary satellites, obervations closest to the analysis time are retained within the (-1h30;+1h30) time range for the assimilation. These details were added in the text : « For non frequent observations at the same location, all observations included in this time range are considered. However for frequent observation types such as radars or radiances from geostationnary satellites, obervations closest to the analysis time are kept within the time range (-1h30;+1h30) in the assimilation. »

**8) Page 4 (line13): "analysed parameters are temperature,. . ." "analysed variables?? are temperature,. . ."** Parameters changed into variables.
**9) Page 5 (Figure 2): Add a), b), c) and d) labels to panels.** Done
**10) Page 6 (Figure 3): Same that in Figure 2.** Done
**11) Page 7 (line 13): "horizontal correlation length-scales are slightly longer". Do the authors refer to the horizontal correlation scales from REANA2? Please improve this sentence.** The sentence was rewritten : « horizontal correlations length-scales are slightly longer in REANA2 than in REANA1 and SOP1 which allows each observation to modify the analysis over a more horizontally extended area. »

Assimilated Data Section:
**12) Page 8 (Table 2): The caption is located very close to the table. Add some additional vertical space between them.** Done
**13) Page 9 (line 9): igher higher.** Corrected

Assimilation Results Section:
**14) Page 20 (line 14): cumulated accumulated** Change made.
**15) Page 20 (Figure 15): Add labels to figure colorbars IOP8 Qualitative Evaluation Section:** Done
**16)Section title: IOP8 Qualitative evaluation IOP8 qualitative evaluation** Modified.
**17) Page 22 (line 12): Gulf ol Lion Gulf of Lion** Done
**18) Page 23 (lines 10-11): Regarding ETS verification score obtained from the daily accumulated precipitation amounts exceeding 50 mm/day, it is stated that ETS scores are better for the 24-48 hour forecast range than for the 00-24 hour forecast period. Could the authors provide some explanation of this result?** The fact that precipitation forecasted at longer ranger are better than those predicted at shorter range suggests that there could be a spin-up effect in the very short forecast ranges that degrades the forecast during the first hours of the forecast. This explanation was added in the text : « This degradation of the short range forecast could originate from a spin-up effect present in the very short ranges of the forecast that degrades the predicted precipitation during the first hours of the forecast. »

---

## Author Response (AR2)

Dear Associate Editor,

I have made all the requested changes. I have also checked the typos and marked the corrections in the new version og the manuscript.

Yours sincerely,

Dr Nadia Fourrié